# Research on the Influence of Geometric Structure Parameters of Eddy Current Testing Probe on Sensor Resolution

**DOI:** 10.3390/s23146610

**Published:** 2023-07-22

**Authors:** Mengmeng Song, Mengwei Li, Shungen Xiao, Jihua Ren

**Affiliations:** 1College of Information, Mechanical and Electrical Engineering, Ningde Normal University, Ningde 352000, China; t1119@ndnu.edu.cn (M.S.); limengwei341622@163.com (M.L.); 2College of Mechanical and Electrical Engineering, Fujian Agriculture and Forestry University, Fuzhou 350002, China; 3Dongguan Xinghuo Gear Co., Ltd., Dongguan 523000, China; sharyren@xhgear.com

**Keywords:** probe, magnetic field, geometry structure, sensor resolution, eddy current testing

## Abstract

To study the influence of the geometric structure of the probe coil on the electromagnetic characteristics of the eddy current probe in the process of eddy current testing, based on the principle of eddy current testing, different probe coil models were established using finite element software. These geometric structure parameters include the difference between the inner and outer radius, thickness, and equivalent radius. The magnetic field distribution around the probe is simulated and analyzed under different parameters, and the detection performance of the probe is judged in combination with the change rate of the magnetic field around the probe coil. The simulation results show that at a closer position, increasing the difference between the inner and outer radii, reducing the thickness, and reducing the equivalent radius are beneficial to improve the resolution of the probe coil. At a far position, reducing the difference between the inner and outer radii, increasing the thickness, and reducing the equivalent radius are beneficial to improve the resolution of the probe coil. At the same time, the accuracy of the simulation data is verified by comparing the theoretical values with the simulated values under different conditions. Therefore, the obtained conclusions can provide a reference and basis for the optimal design of the probe structure.

## 1. Introduction

In practical engineering applications, mechanical equipment generally carries a variety of complex loads, and the various environments in which they continue to work are relatively harsh, including humidity, high pressure, and high temperature. The mechanical properties of this equipment will gradually decrease after a long time of work, and the equipment can even fail. The reason is that in the equipment there are crack defects and corrosion. Non-destructive testing (NDT) technology is significant because it can find these defects in time, which is conducive to ensuring the product quality of this equipment [1,2,3]. The basis of NDT technology is modern science and technology, which is a comprehensive subject. It is usually used to judge the internal or surface structure, physical properties, and state parameters of the tested part. The judgment is usually based on changes in parameters such as sound, light, and magnetism caused by electromagnetic fields, and usually will not destroy the measured object [4]. At present, the commonly used methods of NDT in engineering are mainly the following, such as magnetic particle testing (MT), ultrasonic testing (UT), eddy current testing (ECT), radiographic testing (RT), and penetration testing (PT). Among them, one of the most commonly used NDT techniques is ECT, because of its many characteristics, such as low requirements on the testing surface, no need for contact, fast testing speed, no need for coupling, easy operation, and no radiation to the human body. It plays an important part in some respects because of these advantages, such as testing and evaluating quality and structural integrity, and the detection objects are mainly metal materials, parts, and equipment [5,6,7].

As one of the NDT methods, ECT technology is based on the principle of electromagnetic induction. It uses the magnetic field (MF) energy coupling between the measured object and the probe coil to detect the measured object. This technology is very suitable for testing the integrity of the test piece [8]. When the conductivity, magnetic permeability, and distance between the measured object and the probe change, the corresponding magnetic field intensity will also change. The induced eddy currents induced within it change accordingly, ultimately leading to changes in the output signal of the probe [9,10]. As one of the core components in the eddy current testing system, the eddy current probe undertakes the task of generating the excitation magnetic field and picking up the information of the specimen [11]. The optimization of the probe structure has always been a hot spot in the research of eddy current testing systems.

Cui et al. [12] designed two structural excitation coil models for the detection of ferromagnetic plates, using rectangular coils to induce a directional propagating magnetic field in the plate. Xu et al. [13] carried out research on the optimal design of the far-field eddy current sensor in riveting structure defect detection based on the far-field eddy current detection technology and designed a new type of flat far-field eddy current sensor from two aspects of signal enhancement and magnetic field suppression. The U-shaped probe can generate a uniform magnetic field within a certain space range, and with the increase in the magnetic field strength, the penetration depth becomes deeper [14,15,16]. When the excitation coil is wound in dual excitation mode, the probe has a higher signal-to-noise ratio and is less affected by the lift-off effect [17,18]. Vyroubal conducted theoretical analysis by equating the sensor probe to a transformer model and obtained the relationship between the coil parameters of the probe and the sensitivity and linearity of the probe [19]. Capobianco studied the effects of geometric parameters of the probe coil (inner-to-outer diameter ratio, turns, wire diameter), metal permeability, magnetic core diameter and height, and lift-off height on sensor sensitivity [20]. Tomasz Chady optimized the working frequency and structure of the probe [21]. Young-Kil Shin et al. analyzed a high-performance differential probe structure through finite element simulation and validated the theoretical analysis results through experiments [22]. Chen et al. [23] used ANSYS software to simulate and optimize the size of the rectangular coil and accordingly designed a rectangular-circular probe, which effectively suppressed the impact of lift-off. Gong et al. [24] established a simulation model using the finite element method and analyzed the influence of pulse excitation parameters and excitation coil parameters on the detection sensitivity and resolution by taking the peak value of the differential signal as a feature. Yang et al. [25] proposed and developed a new type of circular eccentric Bobbin probe, which has the functions of axial scanning and eccentric circumferential scanning, and can effectively detect information on small-diameter tube defects. Ahmed et al. [26] proposed the ECECT simulation and hardware design, using the fuzzy logic technique for the development of the new methodology. The depths of the defect coefficients of the probe’s lift-off caused by the coating thickness were measured by using a designed setup. In this result, the ECECT gives an optimum correction for the lift-off, in which the reduction of error is only within 0.1% of its all-out value. Finally, the ECECT is used to measure lift-off in a range of approximately 1 mm to 5 mm, and the performance of the proposed method in non-linear cracks is assessed. Faraj et al. [27] proposed a study on a hybrid giant magneto-resistance/infrared probe to minimize the influence of lift-off for detecting depth defects. The proposed method is verified experimentally, and the result shows that the impact of lift-off noise is highly reduced in the eddy current testing technique and enhances the sensor accuracy. The depth defect error caused by 1 mm lift-off is reduced to 7.20%.

Poletkin et al. [28] derived sets of analytical formulas for the calculation of nine components of magnetic stiffness of corresponding force arising between two current-carrying circular filaments arbitrarily oriented in the space by using Babic’s method and the method of mutual inductance (Kalantarov–Zeitlin’s method). Dziczkowski et al. [29] presented a practical way of using the method of evaluating the metrological properties of eddy current sensors. The idea of the proposed approach consists of employing a mathematical model of an ideal filamentary coil to determine equivalent parameters of the sensor and sensitivity coefficients of tested physical quantities.

It can be seen from the above research that the magnitude and distribution of magnetic induction in eddy current probes are closely related to the resolution and sensitivity of the probe. Most studies focus on the optimization of the probe shape or even the use of dual coils, and there are few studies on the geometric parameters of the probe. Based on this, this paper analyzes the source of affecting the performance of the probe, that is, the magnetic induction intensity, and uses finite element simulation software to construct coil models with different geometric structures. The detection performance of the probe is analyzed by observing the distribution of the magnetic field around the coil, and the theoretical analysis data and simulation data are compared and verified.

## 2. Eddy Current Testing Theory

### 2.1. Eddy Current Testing Principle

One important application of the eddy current (EC) effect is ECT. The principle of ECT is shown in Figure 1. When an alternating current *I*_1_ of a certain frequency is loaded on both ends of the coil shown in the figure, an alternating magnetic field *H*_1_ will be excited around the coil. When the coil is close to the conductive test piece, the alternating magnetic field interacts with the test piece, so the induced current *I*_2_ is generated inside the test piece, and *I*_2_ presents a “vortex” inside the test piece and forms a loop, which is called “eddy current”. According to the principle of electromagnetic induction, *I*_2_ will excite another induced magnetic field *H*_2_. Due to the effect of Lenz’s law, the original magnetic field *H*_1_ and the induced magnetic field *H*_2_ have different directions, so the induced magnetic field will hinder the original magnetic field. The change in the coil impedance signal reflects this effect. By measuring the coil impedance, the detection of the defects of the test piece can be realized, so as to evaluate the performance of the test piece.

The application object of ECT technology is conductive materials. Usually, the bulk density of free charges of metal conductive materials is set to 0, because the movement time of free charges is very short. At this time, the Maxwell equations can be written as:(1)∇×H = (σ+jωε)E
(2)∇×E=−jωμH
(3)∇⋅H=0
(4)∇⋅E=0

Among them, taking the curl of Formula (1) and substituting it into Formula (2), we can obtain:(5)∇×∇×H=(σ+jωε)∇×E

According to the vector relationship, $\nabla \times \nabla \times P=\nabla (\nabla \cdot P)-\nabla^{2}P$, and in Formula (3), ∇⋅H=0; then, we obtain:(6)$$-\nabla^{2}H=(\sigma +j\omega \varepsilon )\nabla \times E$$

Substituting Formula (2) into Formula (6), we obtain:(7)$$\nabla^{2}H-(j\omega \mu \sigma -\omega^{2}\mu \varepsilon )H=0$$

From this, it can be found that the motion form of the electromagnetic field in the medium is a wave. In the actual calculation, it is found that the value of the first term in the brackets of Formula (7) is much larger than the value of the second term. This is because the electrical conductivity of the metal is about 10^7^ Ω^−1^·m^−1^, and the vacuum dielectric constant *ε*_0_ = 8.85 × 10^−12^ F/m; at this time, the ratio of *σ* in the first item to *ωε* in the second item is about 10^−9^, so the second item can be ignored directly, then Formula (7) can be simplified as:(8)$$\nabla^{2}H-j\omega \mu \sigma H=0$$

Similarly, we can also obtain:(9)$$\nabla^{2}E-j\omega \mu \sigma E=0$$
(10)$$\nabla^{2}J-j\omega \mu \sigma J=0$$

Equations (8) to (10) are called electromagnetic penetration equations, which are used to explore the propagation of electromagnetic energy in conductive metals and are also theoretical equations for ECT technology. The meanings of the physical quantities mentioned above are: ∇× is the curl operator; *H* is the magnetic field strength in A/m; *σ* is the conductivity in Ω^−1^·m^−1^; *ε* is the dielectric constant in F/m; *E* is the electric field strength in C/m^2^; *μ* is the magnetic permeability in H/m; ∇⋅ is the divergence operator; and *J* is the current density in A/m^2^.

### 2.2. Impedance Analysis Method

To understand the relationship between the parameters of the probe coil and the properties of the tested object, the researchers proposed an equivalent model, which includes two coil-coupled transformer-coupled mutual-inductance AC circuits. The volt-age effect is observed through the change in coil impedance because there is a similar law between the impedance change and the voltage change. This method is the impedance analysis method, which is widely used in ECT. The equivalent circuit diagram of the coil coupling is shown in Figure 2.

In Figure 2, *R*_1_ and *R*_2_ represent the resistance of the probe coil and the tested piece, respectively; *L*_1_ and *L*_2_ are the inductances of the probe coil and the tested piece, respectively; *M* is the mutual inductance between the probe coil and the tested piece; and *U* is the excitation voltage at both ends of the probe coil.

According to Kirchhoff’s voltage law, the voltage equations in the primary and secondary circuits are:(11)$$\left\{\begin{array}{l} (R_{1}+j\omega L_{1})I_{1}-j\omega MI_{2}=U;\\ (R_{2}+j\omega L_{2})I_{2}-j\omega MI_{1}=0. \end{array}\right.$$

Simultaneously solving the equations in (11), the equivalent impedance of the probe coil can be obtained as:(12)$$Z=\frac{U}{I_{1}}=R_{1}+\frac{{\left(2\pi f\right)}^{2}M^{2}}{R_{2}{}^{2}+{\left(2\pi f\right)}^{2}L_{2}{}^{2}}R^{2}+j[2\pi fL_{1}-\frac{{\left(2\pi f\right)}^{3}M^{2}L^{2}}{R_{2}{}^{2}+{\left(2\pi f\right)}^{2}L_{2}{}^{2}}]$$

By further solving, the equivalent resistance of the real part and the equivalent inductance of the imaginary part of the coil can be obtained as follows:(13)$$\left\{\begin{array}{l} R=R_{1}+\frac{{\left(2\pi f\right)}^{2}M^{2}}{R_{2}{}^{2}+{\left(2\pi f\right)}^{2}L_{2}{}^{2}}R_{2};\\ L=2\pi fL_{1}-\frac{{\left(2\pi f\right)}^{3}M^{2}L_{2}}{R_{2}{}^{2}+{\left(2\pi f\right)}^{2}L_{2}{}^{2}}. \end{array}\right.$$
where the equivalent resistance *R* is a function of the mutual inductance coefficient *M*. It can be observed that *M* increases due to the decrease in the distance between the probe and the tested object, which has nothing to do with whether the tested object is a magnetic material or not. Two effects will affect the equivalent inductance *L*: the magnetostatic effect affects *L*_1_, that is, whether the magnetic material of the test piece is related to *L*_1_; the eddy current effect affects *L*_2_, and the equivalent inductance is oppositely affected by the two effects. Therefore, when the soft magnetic material is used as the tested object, the static magnetostatic effect mainly affects the equivalent inductance in the coil. When the probe is close to the tested object, the equivalent inductance of the probe increases; when non-ferromagnetic material or hard magnetic material is used as the tested object, the eddy current effect mainly affects the equivalent inductance in the coil, and the equivalent inductance of the probe decreases.

### 2.3. Skin Effect

In the related problems of ECT, the attenuated magnetic field induces EC, which will cause the attenuation of the EC inside the conductor specimen. This phenomenon is called the skin effect, that is, the current decays with the increase in depth, and the surface current of the conductor specimen is visibly focused. The penetration depth refers to the distance that the EC penetrates into the conductor. The penetration depth when the EC density decays to 1/e (about 36.8%) of its surface value is defined as the standard penetration depth, also called the skin depth. The formula for calculating the penetration depth of EC is [30,31]:(14)$$\delta =\frac{1}{\sqrt{\pi \sigma \mu }}\frac{1}{\sqrt{f}}$$
where *δ* is the penetration depth in mm; *f* is the frequency of AC current in Hz; *μ* is the magnetic permeability of the conductor in H/m; *σ* is the conductivity of the conductor in S/m.

## 3. Finite Element Model of Eddy Current Detection Probe Coil

### 3.1. The Geometric Structure of the Eddy Current Probe

As the source of an exciting magnetic field and generating eddy current, the eddy current probe coil will directly affect the accuracy and even correctness of eddy current detection. Therefore, it is necessary to explore the influence of the geometric structure parameters of the eddy current probe on the resolution of the sensor, so as to provide a reference and basis for optimizing the probe in practical applications and improving the ability of eddy current flaw detection. The geometric structure parameters of the eddy current probe mainly include the inner diameter *r*_c1_, the outer diameter *r*_c2_, the difference between the inner and outer radii *r*_c2_ − *r*_c1_, the thickness *h*, and the equivalent radius *r*_o_, where *r*_o_ = (*r*_c2_ + *r*_c1_)/2, as shown in Figure 3.

There are many kinds of probe coils, and the multi-turn winding method is selected in this paper. The magnetic induction intensity generated by the coil is distributed on its center line XY, as shown in Figure 4. The inner diameter of the coil is represented by *r*_c1_, the outer diameter is represented by *r*_c2_, the thickness is represented by *h*, and the distance between point X and point Y is represented by *d*. The greater the magnetic induction intensity generated by the coil at point Y and its rate of change, that is, the rate of change, the higher the resolution of the coil here [32].

The resistivity *ρ* and the cross-section *S* of the wire remain consistent in the simulation. The factors that will affect the resistance of the wire are the difference between the inner and outer radius of the coil *r*_c2_ − *r*_c1_, the thickness of the coil *h*, and the equivalent radius of the coil *r*_o_ = (*r*_c1_ + *r*_c2_)/2. Keeping the coil excitation current constant, the difference between the inner and outer radii of the coil, the thickness of the coil, and the equivalent radius of the coil can be expressed as:(15)$$B=\frac{\mu I}{4y}\left[\ln\left(\frac{r_{o}+y+\sqrt{{\left(r_{o}+y\right)}^{2}+h^{2}}}{r_{o}-y+\sqrt{{\left(r_{o}-y\right)}^{2}+h^{2}}}\right)+\frac{r_{o}-y}{\sqrt{{\left(r_{o}-y\right)}^{2}+h^{2}}}-\frac{r_{o}+y}{\sqrt{{\left(r_{o}+y\right)}^{2}+h^{2}}}\right]$$
(16)$$B=\frac{\mu r_{o}{}^{2}I}{2{\left(r_{o}{}^{2}+h^{2}\right)}^{3/2}}$$
(17)$$B=\frac{\mu I}{2h}\left(\frac{d+h}{\sqrt{r_{o}{}^{2}+{\left(d+h\right)}^{2}}}-\frac{d}{\sqrt{r_{o}{}^{2}+d^{2}}}\right)$$
(18)$$\frac{dB}{dr_{o}}=\frac{\mu r_{o}I}{2}\left[\frac{1}{{\left(r_{o}{}^{2}+d^{2}\right)}^{3/2}}-\frac{3r_{o}{}^{2}}{{\left(r_{o}{}^{2}+d^{2}\right)}^{5/2}}\right]$$

Among them, *B* is the magnetic induction intensity, *I* is the excitation current, *r*_o_ is the equivalent radius of the coil, *y* = *r*_c2_ − *r*_c1_ is the difference between the inner and outer radii, *h* is the thickness of the coil, and *d* is the lift-off distance.

### 3.2. Material Selection

Air is selected as the material of the air field, copper is selected as the coil material, and stainless steel is selected as the material of the tested piece. The relative magnetic permeability *μ*_r_, electrical conductivity *σ*, and relative permittivity *ε*_r_ of selected materials are shown in Table 1.

### 3.3. Addition of Physics

There are many modules in COMSOL; here, we select the “Low-Frequency Electromagnetic Fields” module as needed as the physical field of this model. The “Uniform Multi-Turn Coil” is set up in the magnetic field so that the number of turns of the coil and the applied current can be determined. The number of turns of the coil is chosen to be 500, a current signal of 0.05 A is selected as the excitation, and the Dirichlet boundary condition (that is, the magnetic vector potential is zero) is applied to the coil model of the eddy current probe.

### 3.4. Mesh Generation

In the time-varying electromagnetic field, the skin effect will appear in the rotating shaft conductor specimen. At the same time, the magnetic field distribution of defects and probe coil accessories is the key content of this research. Under the premise of not affecting the accuracy of the results, a free triangular mesh is used for the air domain and the probe coil, and the mesh size is selected to be extremely fine. Figure 5 shows the mesh division of the model. Both the eddy current probe and the rotating shaft adopt self-adaptive grids. Since the probe is relatively precise, the grid division is extremely fine, and the volume of the rotating shaft is relatively large, so conventional division can be used. Their specific grid settings are shown in Figure 6a,b as shown.

## 4. Results and Discussion

### 4.1. The Influence of the Difference between the Inner and Outer Radii on the Performance of the Eddy Current Sensor

The coil excitation is kept constant without changing the thickness and equivalent radius. By changing the difference between the inner diameter and the outer diameter of the coil, that is, the difference between the inner and outer radii, the change distribution map of the magnetic induction intensity can be obtained on the center line. When the thickness *h* = 5 mm and the equivalent radius *r*_o_ = 3 mm, the frequency is kept at 1 kHz and the excitation current is kept constant at 0.05 A. When *r*_c2_ − *r*_c1_ are set to 2 mm, 3 mm, 4 mm, and 5 mm, respectively, the results obtained are shown in Figure 7.

It can be seen from Figure 7 that under different differences between the inner and outer radii, as the lift-off distance increases, the magnetic induction intensity first increases and then gradually decreases. In the two intervals of 0.5–1.5 mm and 6–9 mm, when the difference between the inner and outer radii of the coil gradually increases, the magnetic induction intensity generated by the probe coil gradually decreases. In the interval of 1.5–6 mm, when the difference between the inner and outer radii of the coil gradually increases, the magnetic induction intensity generated by the probe coil increases gradually. This change can be intuitively reflected in Figure 8, where the center line gradually transitions from blue-green to orange-red, and then gradually transitions to green and blue.

Since the slope of the curve in Figure 7 represents the change rate of the magnetic induction intensity, which is closely related to the resolution of the probe, it is necessary to explore its change under different differences between the inner and outer radii. Since the change curve of the magnetic induction intensity with lifting distance is nonlinear, by segmenting the curve, two intervals, near and far, are selected to analyze the change rate of magnetic induction intensity. The results are shown in Table 2.

It can be seen from Table 2 that the change rate of the magnetic induction intensity of the coil at a closer position will increase due to the increase in the difference between the inner and outer radii, while at a farther position, the change rate of the magnetic induction intensity will decrease. The rate of change in the magnetic induction is positively related to the resolution of the probe coil. Therefore, it can be seen that increasing the difference between the inner and outer radii can improve the resolution of the probe coil at a closer position, and reducing the difference between the inner and outer radii can improve the resolution of the probe coil at a farther position. This point is more intuitive in the relationship between the rate of change in magnetic induction intensity and the difference between the inner and outer radii, as shown in Figure 9.

In Figure 9, the black curve represents the average magnetic induction intensity change rate in the vicinity, and the red curve represents the far magnetic induction intensity change rate. It can be seen intuitively from the figure that the black curve shows an upward trend, and the red curve shows a downward trend. That is, with the increase in the difference between the inner and outer radii, the resolution of the probe coil is improved at close range, but gradually decreases at a distance.

Under the premise of keeping the assumed conditions consistent, Formulas (15)–(17) mentioned in the previous chapter can be used to calculate magnetic induction. When the difference between the inner and outer radii of the coil is 2 mm and the lift-off distances are 1.5 mm, 3.5 mm, 5.5 mm, and 7.5 mm, the theoretical values of the magnetic induction intensity *B* generated by the coil on its center line are 3.893 mT, 4.386 mT, 3.318 mT, and 1.491 mT, as shown in Table 3.

It can be seen from Table 3 that the simulated value of the magnetic induction on the center line of the coil is compared with the theoretical value, and the ratio of the value is close to 1, that is, it is roughly consistent. Therefore, on the one hand, the accuracy of the simulation data is verified, and on the other hand, the correctness of the law related to the difference between the inner and outer radius of the coil summarized by the simulation data is verified.

### 4.2. The Effect of Thickness on the Performance of Eddy Current Sensor

The excitation loaded on the coil is not changed, and the difference between the inner and outer radii and the equivalent radius remains unchanged. By changing the thickness of the coil, the change distribution of the magnetic induction intensity can be obtained on the center line. When the difference between the inner and outer radii *r*_c2_ − *r*_c1_ = 2 mm and the equivalent radius *r*_o_ = 3 mm, the frequency is kept at 1 kHz and the excitation current remains constant at 0.05 A, and *h* is set to 1 mm, 3 mm, 5 mm, 7 mm and 9 mm, the obtained simulation data results are shown in Figure 10.

It can be seen from Figure 10 that under different probe coil thicknesses, as the lift-off distance increases, the magnetic induction intensity generated by the probe coil first increases and then decreases. Under different coil thicknesses, the position of the peak of the magnetic induction intensity change curve is not the same. As the thickness of the coil increases, the position of the peak value gradually moves backward, and the peak value gradually decreases. At closer distances, as the thickness of the coil increases, the magnetic induction decreases gradually. At a distance, with the increase in the thickness of the coil, the magnetic induction intensity gradually increases. The above changing law can be intuitively reflected in Figure 11.

Since the slope of the curve in Figure 10 represents the change rate of the magnetic induction intensity, which is closely related to the resolution of the probe, it is necessary to explore its change under different differences between the inner and outer radii. Since the change curve of magnetic induction intensity with lifting distance is not linear, the change rate of magnetic induction intensity is analyzed by segmenting the curve and selecting two intervals, near and far. The results are shown in Table 4.

It can be seen from Table 4 that when the thickness of the probe coil is increased, the rate of change in the magnetic induction intensity at a closer position first decreases, then increases, and then decreases again. However, at a farther distance, the rate of change in magnetic induction increases first and then decreases. The rate of change in the magnetic induction is positively related to the resolution of the probe coil. Therefore, it can be seen that reducing the thickness of the coil can improve the resolution of the probe coil in the near area as a whole, and increasing the thickness of the coil can improve the resolution of the probe coil in the distance as a whole. This point is more intuitive in the graph of the relationship between the rate of change in magnetic induction intensity and the thickness of the coil, as shown in Figure 12.

In Figure 12, the black curve represents the average magnetic induction intensity change rate in the vicinity, and the red curve represents the far magnetic induction intensity change rate. It can be seen intuitively from the figure that the black curve shows a downward trend as a whole, and the red curve shows an overall upward trend. That is, on the whole, reducing the thickness of the coil is conducive to improving the resolution of the probe coil in the vicinity, and increasing the thickness is conducive to improving the resolution of the probe coil in the distance.

Under the premise of keeping the assumed conditions consistent, Formulas (15)–(17), mentioned in the previous chapter, can be used to calculate magnetic induction. When the thickness of the coil is 3 mm and the lift-off distances are 1.5 mm, 3.5 mm, 5.5 mm, and 7.5 mm, the theoretical values of the magnetic induction intensity *B* generated by the coil on its center line are 5.299 mT, 4.665 mT, 2.147 mT, and 0.865 mT, as shown in Table 5.

It can be seen from Table 5 that comparing the simulated value of the magnetic induction intensity on the center line of the coil with the theoretical value, the numerical ratio tends to be 1, which is roughly consistent. On the one hand, it verifies the accuracy of the simulation data, and on the other hand, it verifies the correctness of the rules related to the coil thickness summarized through the simulation data.

### 4.3. The Influence of Equivalent Radius on the Performance of Eddy Current Sensor

The excitation applied to the coil remains constant, and the difference between the inner and outer radii and the thickness is kept constant. By changing the equivalent radius of the coil, the change distribution of the magnetic induction intensity is obtained on the center line. When the difference between the inner and outer radii *r*_c2_ − *r*_c1_ = 2 mm and the thickness *h* = 5 mm, the frequency is kept at 1 kHz and the excitation current remains constant at 0.05 A, and *r*_o_ is set to 2 mm, 4 mm, 6 mm, and 8 mm, respectively, and the obtained simulation data results are shown in Figure 13.

It can be seen from Figure 13 that under different equivalent probe coil radii, as the lift-off distance increases, the magnetic induction intensity generated by the probe coil first increases and then decreases. Under different coil equivalent radii, the peak position of the magnetic induction intensity change curve is not the same. With the continuous increase in the equivalent radius of the coil, the position of the peak value gradually moves backward, and the peak value gradually decreases. At a closer distance, the magnetic induction intensity gradually decreases with the increase in the coil equivalent radius as a whole. At a distance, the magnetic induction intensity increases gradually with the increase in the thickness of the coil as a whole. The above variation law can be intuitively reflected in Figure 14.

Since the slope of the curve in Figure 13 represents the change rate of the magnetic induction intensity, which is closely related to the resolution of the probe, it is necessary to explore its change under different differences between the inner and outer radii. Since the change curve of the magnetic induction intensity with lifting distance is not linear, the change rate of magnetic induction intensity is analyzed by segmenting the curve and selecting two intervals, near and far. The results are shown in Table 6.

It can be seen from Table 6 that when the equivalent radius increases, the rate of change in magnetic induction intensity at closer positions will decrease, and at farther positions, the rate of change in the magnetic induction intensity will also gradually decrease. The rate of change in the magnetic induction is positively related to the resolution of the probe coil. Therefore, it can be seen that reducing the equivalent radius of the coil is beneficial to improving the resolution of the probe coil, no matter whether it is near or far away. This point is more intuitive in the graph of the relationship between the rate of change in magnetic induction intensity and the thickness of the coil, as shown in Figure 15.

Under the premise of keeping the assumed conditions consistent, Formulas (15)–(17), mentioned in the previous chapter, can be used to calculate magnetic induction. When the equivalent radius of the coil is 2 mm and the lift-off distances are 1.5 mm, 3.5 mm, 5.5 mm, and 7.5 mm, the theoretical values of the magnetic induction intensity *B* generated by the coil on its center line are 5.299 mT, 4.665 mT, 2.147 Mt, and 0.865 mT, as shown in Table 7.

It can be seen from Table 7 that the simulated value of the magnetic induction intensity on the center line of the coil is compared with the theoretical value, and the numerical ratio is close to 1, which is basically consistent. On the one hand, the accuracy of the simulation data is verified, and on the other hand, the correctness of the laws related to the equivalent radius summarized by the simulation data is verified.

## 5. Conclusions

This paper simulates and analyzes the eddy current detection probe models under several different geometric parameters. Mainly by observing the magnetic induction intensity and its rate of change, exploring the sensor resolution is affected by the geometry parameters of the probe. At the same time, the accuracy of the simulation data is verified by comparing the theoretical values with the simulated values under different conditions. Research shows that:Increasing the difference between the inner and outer radii is beneficial to improving the resolution of the probe coil at a closer position, and reducing the difference between the inner and outer radii is conducive to improving the resolution of the probe coil at a farther position;Reducing the thickness of the coil is conducive to increasing the resolution of the probe coil at close range, and increasing the thickness is conducive to increasing the resolution of the probe coil at a distance;Whether it is near or far away, reducing the equivalent radius of the coil is beneficial to improving the resolution of the probe coil.

## Figures and Tables

**Figure 1 sensors-23-06610-f001:**
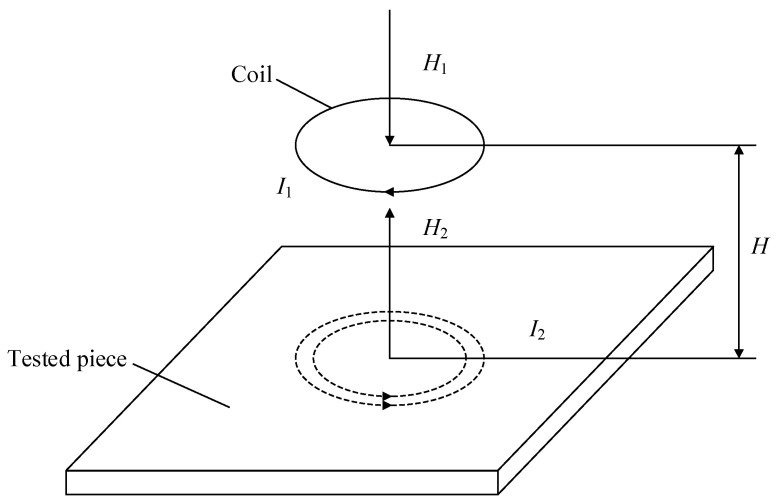
Schematic diagram of eddy current testing.

**Figure 2 sensors-23-06610-f002:**
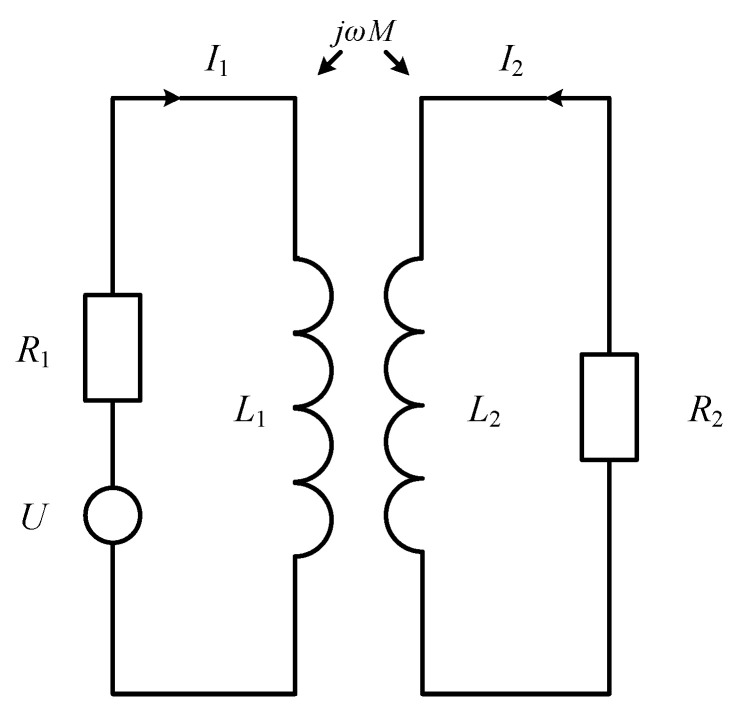
Equivalent circuit diagram.

**Figure 3 sensors-23-06610-f003:**
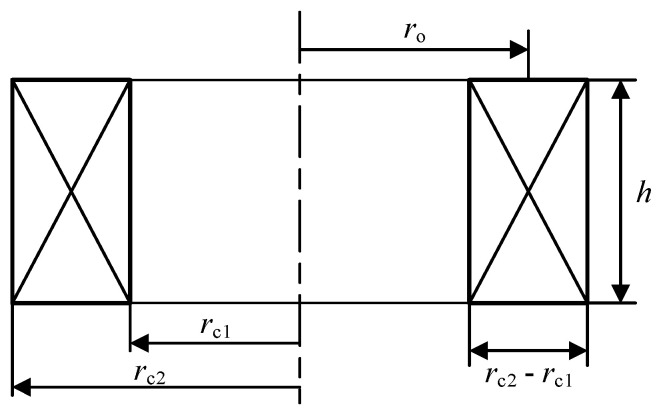
Probe geometry parameter diagram.

**Figure 4 sensors-23-06610-f004:**
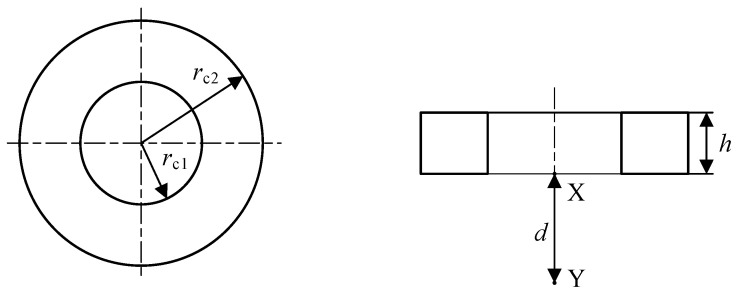
Probe coil schematic.

**Figure 5 sensors-23-06610-f005:**
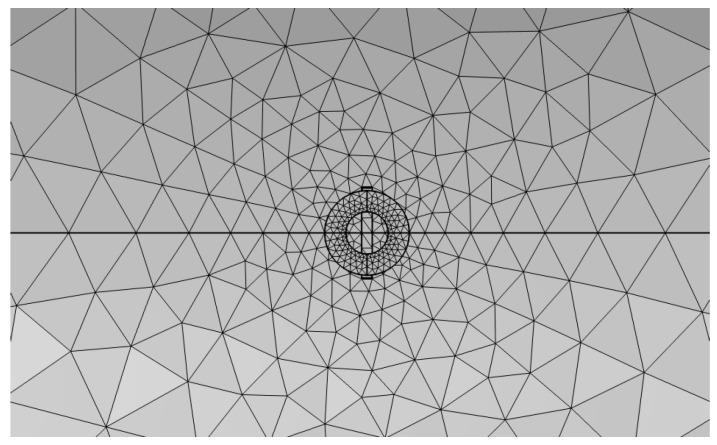
Mesh division diagram of the model (vertical view).

**Figure 6 sensors-23-06610-f006:**
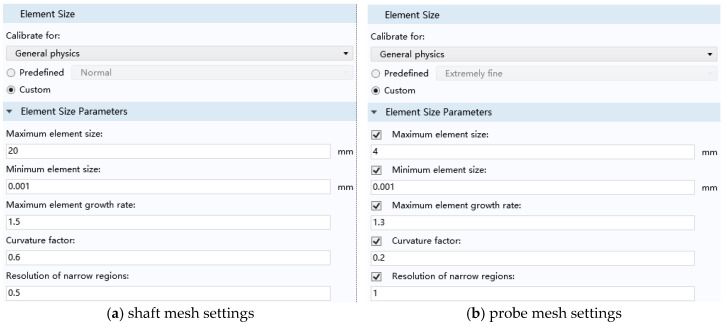
Shaft and probe mesh settings.

**Figure 7 sensors-23-06610-f007:**
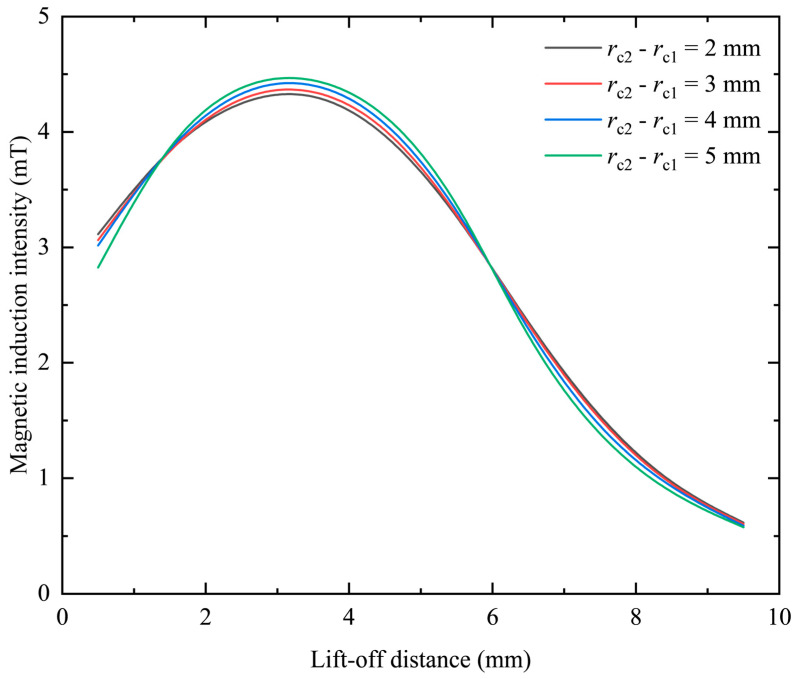
The change curve of magnetic induction intensity when the differences between the inner and outer radii of the coil are different.

**Figure 8 sensors-23-06610-f008:**
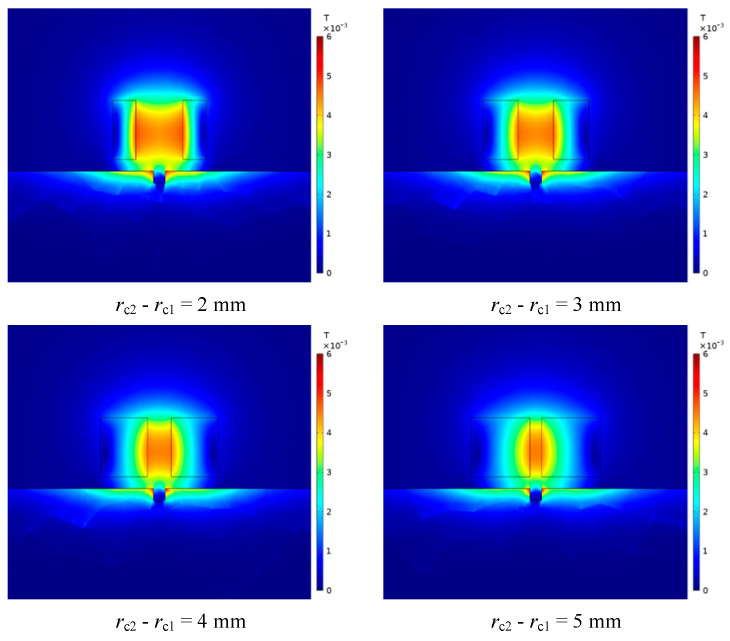
The distribution of magnetic induction intensity when the differences between the inner and outer radii of the coil are different.

**Figure 9 sensors-23-06610-f009:**
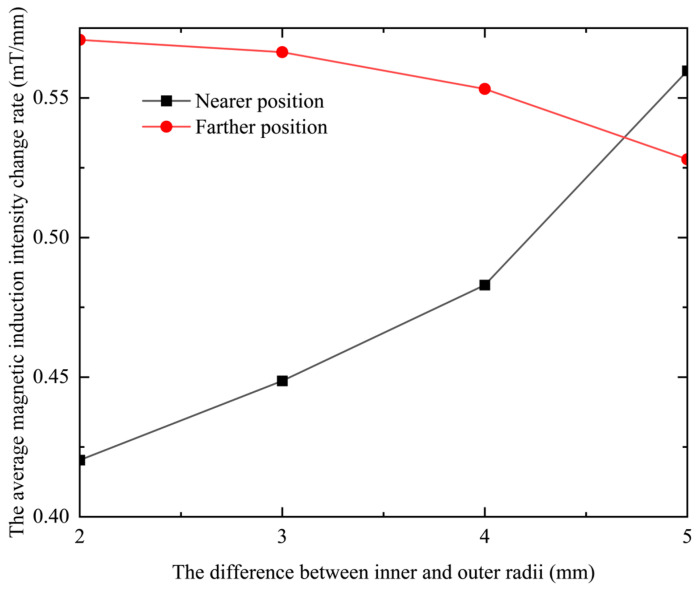
The relationship between the change rate of the average magnetic induction intensity and the difference between the inner and outer radii.

**Figure 10 sensors-23-06610-f010:**
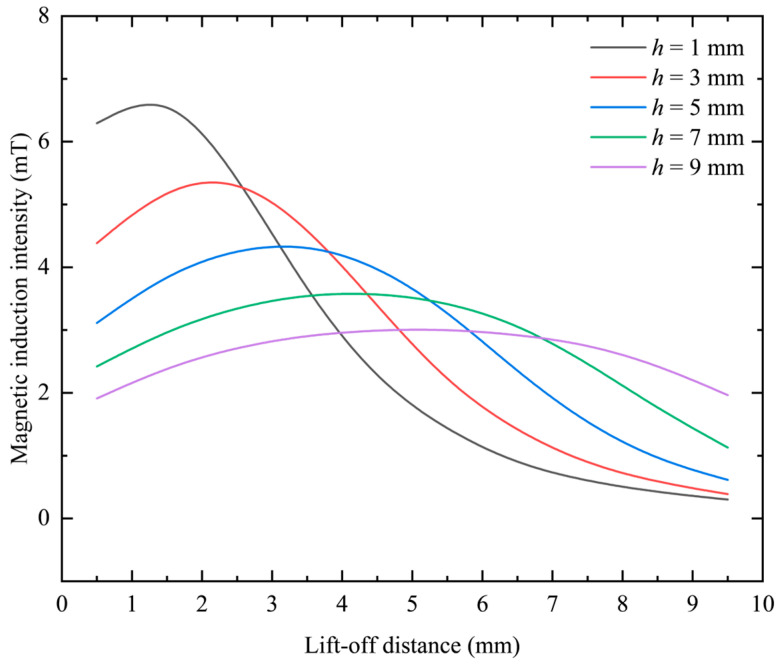
The influence of probe thickness on magnetic induction intensity.

**Figure 11 sensors-23-06610-f011:**
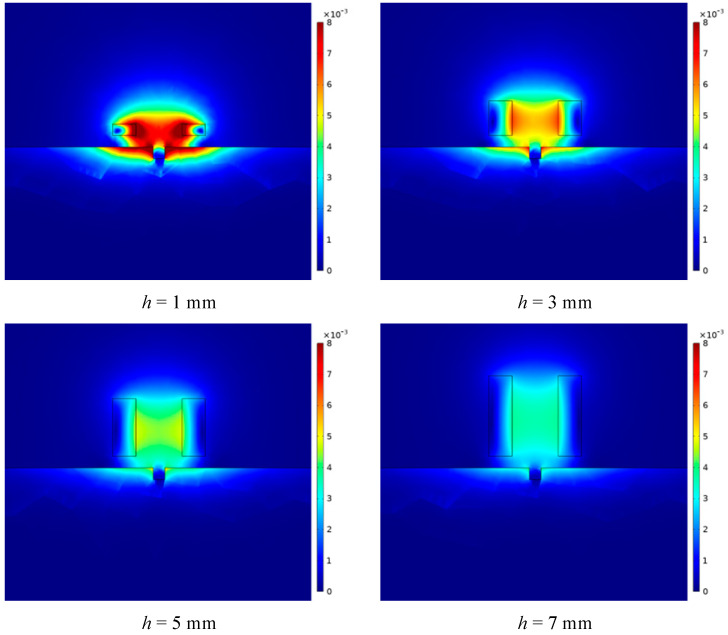
The distribution of magnetic induction intensity when the thicknesses of the coil are different.

**Figure 12 sensors-23-06610-f012:**
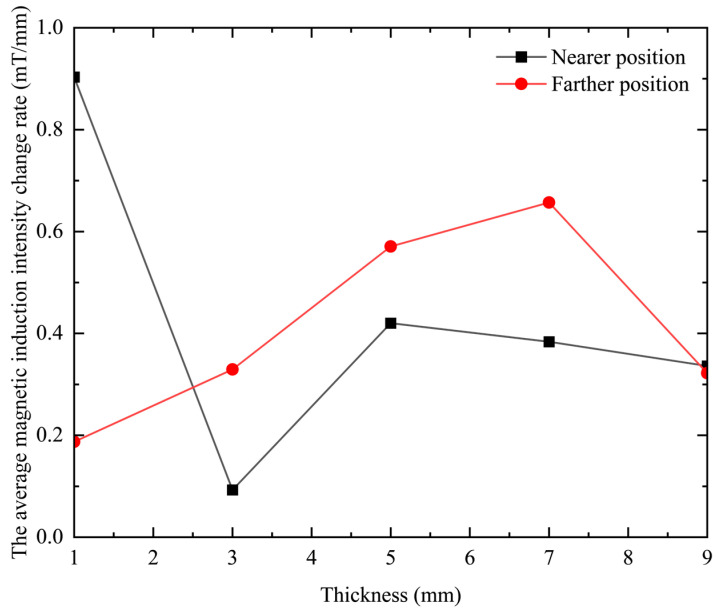
The relationship between the average magnetic induction intensity change rate and the thickness of the coil.

**Figure 13 sensors-23-06610-f013:**
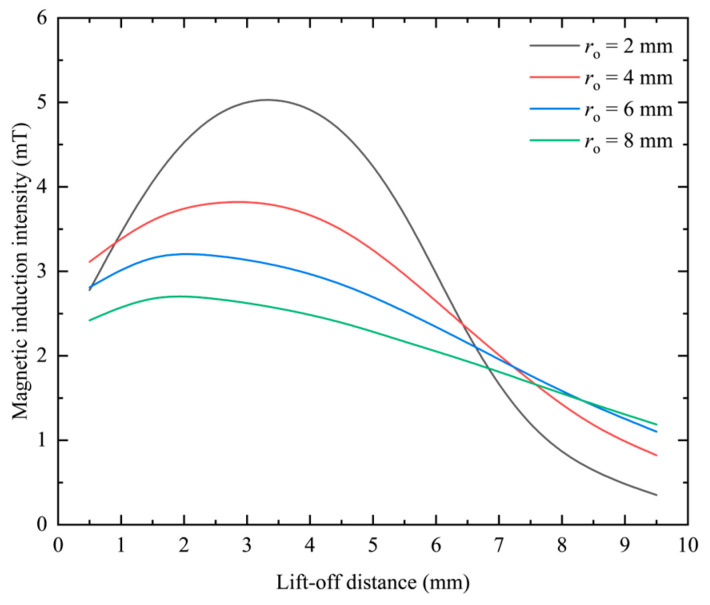
The influence of coil equivalent radius on magnetic induction.

**Figure 14 sensors-23-06610-f014:**
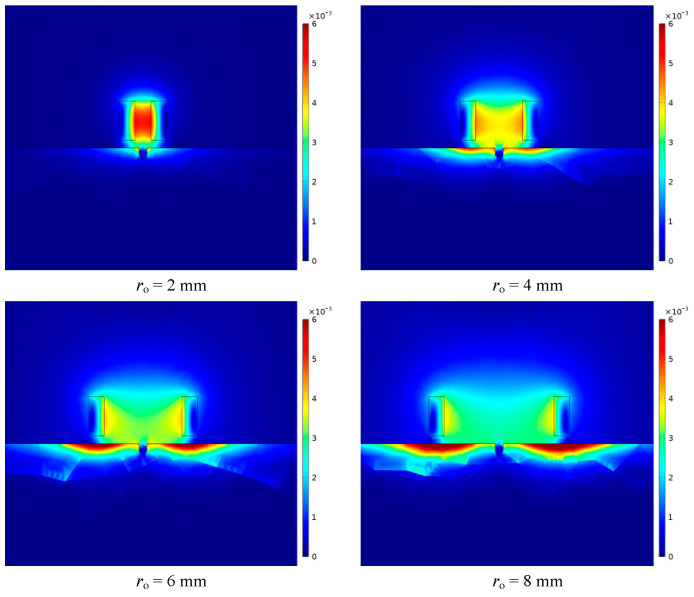
The distribution of magnetic induction intensity when the equivalent radii of the coil are different.

**Figure 15 sensors-23-06610-f015:**
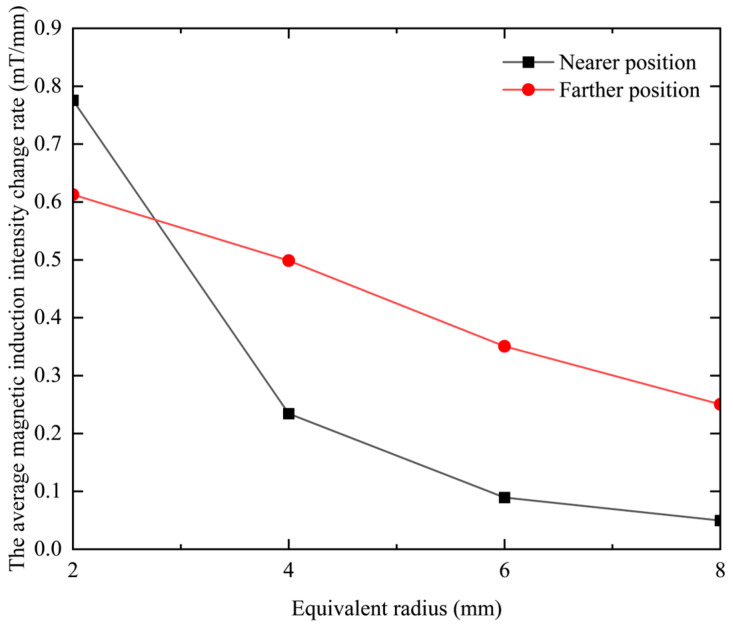
The relationship between the change rate of the average magnetic induction intensity and the equivalent radius of the coil.

**Table 1 sensors-23-06610-t001:** Material parameters.

Material	*μ* _r_	*σ*/(S•m^−1^)	*ε* _r_
Air	1	0	1
Copper	1	5.998 × 10^7^	1
Stainless steel	100	1.137 × 10^6^	1

**Table 2 sensors-23-06610-t002:** The rate of change in the magnetic induction intensity at different differences between the inner and outer radii.

The Difference between Inner and Outer Radii *r*_c2_ − *r*_c1_ (mm)	The Average Magnetic Field Change Rate When the Lift-Off Distance Is 0.5–3.5 mm (mT/mm)	The Average Magnetic Field Change Rate When the Lift-Off Distance Is 6.5–9.5 mm (mT/mm)
2	0.4203	0.5708
3	0.4487	0.5664
4	0.4830	0.5532
5	0.5597	0.5280

**Table 3 sensors-23-06610-t003:** Comparison of simulated and theoretical values of magnetic induction on the center line of the coil when the difference between the inner and outer radii is 2 mm.

Lift-Off Distance (mm)	Theoretical Value (mT)	COMSOL (mT)	Ratio
1.5	3.893	3.901	1.002
3.5	4.386	4.375	0.997
5.5	3.318	3.308	0.996
7.5	1.491	1.495	1.003

**Table 4 sensors-23-06610-t004:** The rate of change in magnetic induction at different probe coil thicknesses.

Thickness *h* (mm)	The Average Magnetic Field Change Rate When the Lift-Off Distance Is 0.5–3.5 mm (mT/mm)	The Average Magnetic Field Change Rate When the Lift-Off Distance Is 6.5–9.5 mm (mT/mm)
1	0.9033	0.1872
3	0.0927	0.3293
5	0.4203	0.5708
7	0.3837	0.6570
9	0.3363	0.3220

**Table 5 sensors-23-06610-t005:** Comparison of simulated value and theoretical value of magnetic induction on the center line of the coil when the thickness is 3 mm.

Lift-Off Distance (mm)	Theoretical Value (mT)	COMSOL (mT)	Ratio
1.5	5.299	5.302	1.001
3.5	4.665	4.661	0.999
5.5	2.147	2.153	1.003
7.5	0.865	0.863	0.998

**Table 6 sensors-23-06610-t006:** The rate of change in magnetic induction at different probe coil equivalent radii.

Equivalent Radius *r*_o_ (mm)	The Average Magnetic Field Change Rate When the Lift-Off Distance Is 0.5–3.5 mm (mT/mm)	The Average Magnetic Field Change Rate When the Lift-Off Distance Is 6.5–9.5 mm (mT/mm)
2	0.7757	0.6127
4	0.2343	0.4986
6	0.0893	0.3507
8	0.0497	0.2503

**Table 7 sensors-23-06610-t007:** Comparison of simulated value and theoretical value of magnetic induction intensity on the center line of the coil when the equivalent radius is 2 mm.

Lift-Off Distance (mm)	Theoretical Value (mT)	COMSOL (mT)	Ratio
1.5	4.152	4.157	1.001
3.5	5.089	5.104	1.003
5.5	3.768	3.764	0.999
7.5	1.092	1.095	1.003

## Data Availability

The datasets generated and analyzed during the current study are not publicly available due the data also forming part of an ongoing study, but are available from the corresponding author on reasonable request.

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
