# Peer review of "Research on the Influence of Geometric Structure Parameters of Eddy Current Testing Probe on Sensor Resolution"

_sensors, 2023, doi:10.3390/s23146610_

Round 1
Reviewer 1 Report
1. In the abstract, the phrase "probe coil models were established by using finite element software" can be rephrased as "probe coil models were established using finite element software" for better clarity and flow.
2. In the abstract, the sentence "The distribution of the magnetic field around the probe under different parameters is simulated and analyzed" could be modified to "The distribution of the magnetic field around the probe is simulated and analyzed under different parameters" to enhance readability.
3. Introduction: Consider providing a clear and concise introduction that sets the context for the study. Start by briefly explaining the importance and applications of eddy current testing. Then, highlight the significance of probe coil geometry in determining the resolution of the eddy current probe. Provide a brief overview of previous research in this area and identify any gaps or limitations that your study aims to address. You may check these refernces
A. https://doi.org/10.3390/s18072108
B. https://journals.sagepub.com/doi/10.1177/0020294019827336
4. Section 3: It would be beneficial to provide more details about the simulation model used for establishing the probe coil models. Specifically, describe the specific finite element software employed, the parameters used in the simulation, and any assumptions made during the modeling process.
5. Eddy Current Sensor Parameters: Consider including a table in the paper that provides a comprehensive overview of the eddy current sensor parameters used in the simulation. This table can include details such as the dimensions of the probe coil, material properties, operating frequency, and any other relevant parameters.
6. Update the References: Verify that all the references cited in the paper are up-to-date and from reputable sources. It is essential to ensure that the cited references align with the current state of research in the field. Additionally, cross-check the references with the citation style guidelines of the target journal or publication to ensure consistency and accuracy.
7. It is important to address the issues related to the English language in the paper. Poor grammar, sentence structure, and word choice can hinder the clarity and readability of the manuscript.
Author Response
We seriously revise the contents of this paper according to the comments. We appreciate for reviewers’ warm work earnestly, and hope that the corrections will meet with approval. If they are inappropriate, we will be very glad to receive your opinions and will make further modification to this paper. Once again, thank the esteemed editor and reviewers for your hard work.

Reviewer 2 Report
The paper presents the influence of the geometry of the eddy current coil on the properties of the magnetic field. Parameters such as the height and width of the coil and its equivalent radius were analyzed.
Unfortunately, the manuscript is written in a way that is unclear to me, and for this reason I have many comments about it.
Minor remarks
1) The paper uses an ambiguous way of citation. Sometimes they are superscripts, and sometimes a text-sized font was used.
2) The authors' names mentioned in the text and in the References differ from each other, e.g. "Youang kil Shin et al. analyzed ... [22]". The correct name of the author is slightly different. Also, in the References section, the dash "-" is omitted from many names.
3) The literature review in the Introduction is solid and contains many papers. However, it should be slightly supplemented by citing articles that used a equivalent radius (K. Poletkin, S. Babic, L. Dziczkowski) so that the reader can become familiar with this approach.
4) The authors use the phrase “the formula mentioned in the previous chapter” many times. Which formula is it? Always specify the expression number.
5) Is the coil placed over some conductive material? What were its parameters, e.g. electrical conductivity, magnetic permeability? Have both magnetic and non-magnetic materials been studied?
6) Formulas (15)-(18) are the only expressions developed in the work, so they certainly require a more detailed discussion.
Major comments
7) I don't know what is the novelty of the manuscript. Why perform the calculations presented in the paper when you can use the finite element method?
8) It is difficult to perform an analysis of the results obtained on the basis of graphs containing only 4-5 points (for example, Figs. 7, 10, 13).
9) Verification is the weak point of the job. No experiments were performed. Nothing is known about the parameters of the FEM model used - boundary conditions, shape of elements, number of mesh elements. Not knowing the values of such important parameters - I cannot verify the presented results.
10) "In this paper, the eddy current detection probe models under different geometric parameters are established." - this sentence is too strong. In my opinion, the article presents only 4 expressions (15)-(18) concerning induction B and the results of simulations in the Comsol program. At the same time, I think that the mathematical contribution in the work is too small.
11) "reducing the equivalent radius of the coil is beneficial to improving the resolution of the probe coil no matter it is near or far away." – this should be clarified, because in the literature there are opinions that the increase in the equivalent radius of the coil is beneficial and desirable, e.g. in
L. Dziczkowski, G. Tytko, Evaluation of the Properties of Eddy Current Sensors Based on Their Equivalent Parameters, Sensors 23 (6), 3267, 2023.
12) In my opinion, the final conclusions are questionable and useless. I don't know how they can be used for probe design. The terms "close" and "far" are imprecise. It seems to me that conclusions 1 and 3 do not coincide - the phrase from conclusion 1 "reducing the difference between the inner and outer radii is conducive to improving the resolution of the probe coil at a farther position" refers to the case when decreasing the equivalent radius. Meanwhile, proposal 3 shows that increasing the equivalent radius is beneficial.
In summary: simple simulations in Comsol, 4 expressions and 3 conclusions that I don't know how to apply are far too few to consider that the manuscript contributes to the eddy current testing literature.
Author Response

(The authors gave the same response as above.)

Round 2
Reviewer 2 Report
The authors have responded to all the comments and have improved the manuscript.
Author Response

(The authors gave the same response as above.)
